# Effect of Different Formulations and Storage on the Physicochemical, Microbiological, and Organoleptic Characteristics of *Dovyalis caffra* Fruit Yogurt

**DOI:** 10.3390/foods13244102

**Published:** 2024-12-18

**Authors:** Daniel Mwangi Waweru, Joshua Mbaabu Arimi, Eunice Marete, Niamh Harbourne, Jean-Christophe Jacquier

**Affiliations:** 1Department of Food Science and Technology, Meru University of Science and Technology, Meru, Kenya; damwas31@yahoo.com (D.M.W.); jarimi@must.ac.ke (J.M.A.); 2Department of Physical Sciences, Meru University of Science and Technology, Meru, Kenya; emarete@must.ac.ke; 3Institute of Food and Health, School of Agriculture and Food Science, University College Dublin, Belfield, Dublin, Ireland; niamh.harbourne@ucd.ie

**Keywords:** *Dovyalis caffra*, yogurt, physicochemical, microbiological, organoleptic

## Abstract

This study investigated the effect of incorporating up to 15% (*w*/*w*) *Dovyalis caffra* fruit pulp into cow milk yogurt. Monitoring the physico-chemical, microbiological, and organoleptic properties of these formulations was performed weekly during refrigerated (4 °C) storage for 21 days. Compared to the control, formulations with added pulp recorded enhanced contents of ascorbic acid, total polyphenols, fiber, total titratable acidity (TTA), and yellowness, which is in line with increasing fruit pulp. This, however, corresponded to significantly lower contents in fat, protein, viscosity, pH, and overall sensory acceptability. Pulp incorporation had no significant effect on moisture, ash, and water-holding capacity (WHC) of formulations. Storage for up to 21 days indicated a significant increase in TTA and a corresponding decrease in pH but no significant change in WHC and viscosity of formulations. There was no detection of coliforms, yeasts, and molds in all samples throughout the storage period. This study demonstrates that *Dovyalis caffra* fruit has significant potential in the formulation of nutritious, desirable, and shelf-stable fruit-based yogurts. Further product optimization is, however, recommended to maximize the organoleptic quality of the formulations.

## 1. Introduction

*Dovyalis caffra* is one of Africa’s indigenous fruits that remains highly underutilized among many other local species [1]. Despite such underutilization, the fruit has previously been demonstrated to exhibit considerable nutritional, health, and physicochemical potential for application in food systems [2]. It is rich in polyphenols, carotenoids, vitamins, minerals, amino acids, and sugars, among other vital compounds [3,4,5,6]. It is also high in acidity and pectin levels and, thus, bears the potential to form stable gels in a food matrix [2]. The bright yellow color of the fruit [1] further indicates its potential for use in color enhancement of other foods. Clearly, the creation and expansion of ways to harness *Dovyalis caffra’s* fruit potential in the food system is warranted. One strategic way of introducing a less known or underutilized food material to the market is by incorporating it in a popular food product that acts as a carrier. Yogurt may, for example, be considered a potentially suitable carrier for an underutilized fruit due to its extensive ingredient compatibility range, straightforwardness of processing, universal prevalence, and socio-economic viability.

By definition, yogurt is the product developed by lactic acid fermentation of milk through the action of *Lactobacillus delbrueckii* ssp. *bulgaricus* and *Streptococcus thermophilus* on milk lactose [7]. It is one of the most popular fermented dairy products that is linked to various health benefits associated with high calcium levels and bioactivity linked to the use of live microorganisms in the product [8]. Fruit yogurt, on the other hand, is defined as the product obtained by the addition of fruits and their nectars, jams, marmalade, fruit jellies, fruit syrups, and concentrated fruit drinks to yogurt or cultured pasteurized milk [8]. The addition of fruit components to yogurt accentuates the health image of the product through the provision of diverse nutrients and phytochemicals that are otherwise limited or absent in dairy products [9,10]. Additionally, such fruit components have been reported to have varying impacts on the overall physicochemical, sensory, and microbiological characteristics of the yogurts [11,12,13].

Recently, a study of set yogurt incorporated with orange fruit pulp ranging from 0 to 4% reported a significant increase in the content of fat, protein, total soluble solids, moisture, ascorbic acid, antioxidant activity, and syneresis of product with pulp addition [14]. In the said study, 4% pulp proved to be the most optimal, with the product exhibiting a desired range of pH, total soluble solids, and titratable acidity during refrigerated (4 °C) storage for three weeks.

A study involving osmo-dihydro-frozen fruits [8] indicated that 10% apple or 13% strawberry in yogurt formulations yielded the best quality products during processing. In this study, fruit yogurts with higher fruit percentages recorded higher preference in terms of textural/mouth-feel sensory attributes, while storage led to a significant decrease in pH, syneresis, and organoleptic scores regarding both taste and texture. However, the same study further showed that the apple fruit yogurt was yeast and mold-free, and its coliform levels were eliminated within the first week of storage, while the strawberry yogurt’s yeasts and molds increased during the first week of storage before being reduced again after 14 days, while coliform contamination remained low but persisted for up to 2 weeks of storage.

In a study by Kamber et al. [15], fruit yogurt formulations comprising 20% banana, strawberry, peach, and apricot pulps showed significant differences in the physicochemical and microbiological characteristics between fruit yogurts and plain yogurt during storage for 7 days at 4 °C. That study showed strawberry yogurt had the highest total titratable acidity (TTA) and counts of *Lactobacillus bulgaricus,* while peach had the lowest pH and highest count of *Streptococcus thermophilus*. Plain yogurt had the highest yeast count, and banana yogurt had the lowest mold count. Sensory properties were not significantly affected by the type of fruit. All these studies are a clear demonstration that the addition of fruit components to yogurt has varying effects on the physicochemical, sensory, and microbiological properties of the product.

Specific studies involving *Dovyalis caffra* in relation to such aspects have not, however, been reported. The current study, therefore, aims to determine the effect of different pulp ratios on the physicochemical, sensory, and microbiological properties of *Dovyalis caffra* fruit yogurt formulations and also determine the effect of a 21-day refrigerated storage (4 °C) period on the physicochemical, sensory, and microbiological properties of the fruit yogurt formulations. This pioneering study is expected to create a potential way of utilizing the highly underutilized *Dovyalis caffra* fruit through the production of novel value-added yogurts with desirable sensory, physicochemical, and microbiological properties.

## 2. Materials and Methods

### 2.1. Materials

Fully ripe *Dovyalis caffra* fruits were harvested in January 2022 from Lanet town in Nakuru County, Kenya. The fruit was packed in corrugated fiberboard boxes and transported by road to the Food Science Laboratories of Meru University of Science and Technology, Kenya, for analysis. Fresh cow milk was purchased from a local dairy farmer in Meru County. Starter culture (*Lactobacillus bulgaricus* and *Streptococcus thermophilus*) was obtained from Chr Hansen, Denmark, while sugar and skimmed milk powder were obtained from a retail shop in Meru town. Analytical-grade chemical reagents were purchased from Sigma-Aldrich (St. Louis, Mo, USA) through Kobian Kenya Ltd (Nairobi, Kenya).

### 2.2. Fruit Pulp Preparation

On reception at the laboratory, *Dovyalis caffra* fruits were sorted, cleaned in cold water, rinsed, and drained on racks to surface dryness. The fruits were then randomly aliquoted into three portions prior to processing through a pulper-finisher (Type MK-1, Larsen & Toubro Ltd., Mumbai, India) to obtain fruit pulp free of skin and seeds. The extracted pulp was pasteurized by the method of Matter et al. [11] with slight modification. Heating of *Dovyalis caffra* fruit pulp was performed at 95 °C for 15 min using stainless steel containers immersed in a water bath. This was followed by cooling to room temperature (25 °C) prior to utilization in fruit yogurt formulation.

### 2.3. Fruit Yogurt Formulation

Plain milk yogurt was processed according to the standard protocol used in the Dairy section of the Food Science laboratories of Meru University of Science and Technology, Kenya. Briefly, 2.5% skimmed milk powder and 6.5% sugar were dissolved in 3 L batches of milk with concurrent pasteurization of such mixture at 90 °C for 30 min. The sugar and skimmed milk powder were added after the milk attained a temperature of at least 40 °C. After heating, the pasteurized mixture was then cooled to 45 °C, followed by inoculation with 0.006% of starter culture (No. YFL811) constituting *Lactobacillus bulgaricus + Streptococcus thermophilus* (1:1), (Chr Hansen, Horsholm, Denmark) and subsequent incubation in a cheese vat at the said temperature for 6 h. The resultant plain yogurt was subsequently refrigerated at 4 °C overnight (12 h) prior to its utilization in fruit yogurt formulation.

Into the plain yogurt, different proportions of pasteurized *Dovyalis caffra* fruit pulp were incorporated to make formulations with 0%, 5%, 10%, and 15% fruit components by weight. Pulp addition and mixing were carried out at the stirring stage of yogurt; hence, no extra treatment was necessitated. Standard stirring protocol prevailed for all formulations, including the control sample D0.

These fruit yogurt formulations (Figure 1) are herein denoted as (D-0), (D-5), (D-10), and (D-15,) respectively. Finally, the formulations were packed in 250 mL sterilized plastic bottles, sealed, and kept under refrigeration at 4 °C prior to relevant physicochemical, microbiological, and sensorial determinations.

### 2.4. Proximate Analysis

The moisture content of yogurt formulations was determined according to the Association of Official Analytical Chemists (AOAC, 1995) [16], Method 925.10–32.1.03. Sample weight loss after complete drying at 105 °C was expressed as percentage moisture content.

The percentage nitrogen content of yogurt formulations was analyzed by the Kjeldahl method, as illustrated in the AOAC, 1995 [16], Method 20.87–32.1.22. Protein % = Nitrogen × protein factor. (Protein factor = 6.38).

The contents of fat, ash, and fiber of yogurt formulations were also determined according to the AOAC, 1995 [16], Methods; 920.85–32.1.13, 923.03–32.1.05 and 920.86–32.1.15, respectively.

The total carbohydrate content of yogurt samples was determined by the difference method as follows: % total digestible carbohydrates = 100 − [total ash + fiber + fat + protein + moisture].

### 2.5. Ascorbic Acid and Total Polyphenols

The ascorbic acid content of yogurt samples was determined by the 2,6-dichloroindophenol titrimetric method, as described by Mazumdar, 2009 [17], while the total phenolic content (TPC) was determined by the Folin–Ciocalteu method with a slight modification of the procedure described by Singleton et al. [18]. The total phenolic content was calculated using a gallic acid standard curve, and results were expressed as mg of gallic acid equivalents (GAE) per 100 g.

### 2.6. Physicochemical Properties

The pH of yogurt formulations was measured using a calibrated digital pH meter (Hanna-instruments, Woonsocket, RI, USA).

The TTA of yogurt formulations was determined by sample titration using 0.1 N NaOH and phenolphthalein as an indicator. %TTA was expressed as % lactic acid, *w*/*w* [8].

The water-holding capacity of yogurt formulations was measured by the whey expulsion method following a slight modification of the procedure described by Matter et al. [11]. Firstly, 2 g of yogurt sample was weighed in a vial and then centrifuged at a Relative Centrifugal Force (rcf) of 1968 g for 30 min at 10 °C. The supernatant was then poured out and weighed. WHC was calculated as the difference in weight between the original sample and the residual sample after whey expulsion. Results were expressed as a percentage.

Apparent viscosity was determined by the use of a concentric cylinder rheometer (RheolabQC—Anton Paar, Graz, Austria) with a slight modification of the method described by Renan et al. [19]. About 10 mL of yogurt sample was placed in the concentric cylinder rheometer cup before mounting onto the rheometer and conducting measurement at a shear rate of 64/s for 2 min. Apparent viscosity was automatically computed using software (RheoPlus/32), with readings taken after 12 s (10th point) of the shear stress–shear rate spectrum, displayed on the digital output of the rheometer. The readings were expressed in Pa·s.

The color of various yogurt samples was measured using a colorimeter (Konica, Minolta-Spectrophotometer-CM-700D, Tokyo, Japan) with geometry CR-A33c and using illuminant D65, based on the L*, a*, and b* coordinate system, as described by Dimitrellou et al. [20].

### 2.7. Microbiological Analysis

All media used for the microbiological analysis of yogurts were prepared in strict adherence to the respective manufacturer’s instructions. Sample preparation entailed weighing 11 g of yogurt sample and homogenization of sample in 99 mL of sterile peptone water (magnetic stirrer at 40 °C for 10 min) to form 1:10 dilution.

Yeast and mold counts were determined by pour-plating 1 mL of prepared sample dilution on acidified potato dextrose agar (PDA). Plate incubation was performed at 25 °C for 5 days. Results were expressed as colony-forming units per gram (cfu/g) of the sample [21].

Coliform counts of yogurt formulations were determined by pour-plating 1 mL of prepared sample dilution on violet red bile agar (VRBA). Plate incubation was performed at 30 °C for 24 h; typical red colonies (>0.5 mm) were considered coliforms. Coliform counts were expressed as colony-forming units per gram (cfu/g) of the sample [21].

### 2.8. Sensory Analysis

Sensory analysis of yogurt formulations was conducted on day 1, day 7, day 14, and day 21 of storage by a team of 28 untrained panelists [15]. A nine-point hedonic scale (9 = Like extremely; 1 = Dislike extremely) was used to score the products’ organoleptic attributes of color, taste, texture, and overall acceptability.

### 2.9. Statistical Analysis

The overall study entailing yogurt formulations and their characterization was repeated twice (duplicate). For each specific experiment, all samples were analyzed in triplicate sets [22]. Obtained data were subjected to a one-way analysis of variance (ANOVA) coupled with Tukey’s honest significant difference (HSD) test for the determination of significant variations among the means [22]. A significant difference was established at *p* < 0.05 [22]. All statistical computations were carried out using GenStat data analysis software—14th edition (VSN International, Hemel Hempstead, UK).

## 3. Results and Discussion

### 3.1. Basic Composition of Dovyalis caffra Fruit Yogurt

The basic chemical composition of freshly prepared (1-day-old) *Dovyalis caffra* fruit yogurt is presented in Table 1. From general observation, a 5% increase in pulp ratio led to significantly higher contents in fiber, carbohydrate, ascorbic acid, and total polyphenols among respective formulations. With regard to protein and fat, formulation D-15 significantly recorded the lowest content in comparison to the control, while the rest of the formulations registered no significant difference. Overall, all formulations recorded no significant difference in their moisture and ash contents. The lower fat and protein contents of yogurt formulations with increased *Dovyalis caffra* fruit are similar to that reported in carrot juice yogurt formulations [21] and are indicative of the typically low levels of these components in fruits and vegetables. Similarly, the increasing trend exhibited in the fiber, carbohydrate, ascorbic acid, and total polyphenols of yogurt formulations with higher *Dovyalis caffra* fruit pulp ratios is indicative of the typically high levels of these components in fruits compared to dairy products. These results agree fairly well with the findings of Matter et al. [11] in their study on papaya and cactus fruit yogurt formulations. Overall, the addition of fruit components to plain milk yogurts is shown to significantly improve the nutritional and phytochemical composition of such products and, hence, their health image.

### 3.2. Physicochemical Properties and Their Evolution During Storage of Dovyalis caffra Fruit Yogurt

The physicochemical properties of freshly prepared (1-day-old) *Dovyalis caffra* fruit yogurt formulations are presented in Table 2. As shown in the table, a consecutive increase of 5% in the amount of incorporated fruit pulp produced yogurt formulations with significantly lower pH and corresponding higher TTA among all the products. This could be attributed to the increased concentration of organic acid per unit volume of product, considering that *Dovyalis caffra* fruit is typically very acidic [23]. Such variation in pH and TTA has similarly been reported in cactus pear yogurt formulations [11].

As indicated in the same table, refrigerated storage (4 °C) led to a significant decrease in the pH and a corresponding increase in TTA of all yogurt formulations after the 7th and 14th day of storage period. Such changes are indicative of increased acidity resulting from the activity of the Lactic acid bacteria used as starter culture. These results agree fairly well with the findings of Matter et al. [11] and Aly et al. [24] on papaya pulp yogurt and carrot juice yogurt formulations, respectively.

Though not significant, the incorporation of *Dovyalis caffra* fruit pulp into plain milk yogurt was shown to result in numerically lower apparent viscosities of all freshly prepared formulations in comparison to the control (Table 2) in line with the findings of Kermiche et al. [25] in the case of cantaloupe puree yogurts. As further demonstrated in Table 2, a storage period of 21 days did not show any significant effect on the viscosity of all *Dovyalis caffra* yogurt formulations. This is an indication of stability in the various yogurt microstructures, as may be affirmed by the report of Kermiche et al. [25] regarding stored cantaloupe puree yogurt formulations.

The water-holding capacity of freshly made (1-day-old) *Dovyalis caffra* fruit yogurt formulations is indicated in Table 2. As shown in the table, there was no significant variation in the WHC of all formulations. Similar results have been reported in grape juice yogurt formulations [20] and papaya yogurt formulations [11]. The findings of Dimitrellou et al. [20] in stored grape juice yogurt formulations further affirm the insignificant change in WHC of *Dovyalis caffra* fruit yogurt formulations during storage for 21 days (Table 2).

With regard to the color of freshly prepared (1 day old) *Dovyalis caffra* fruit yogurt formulations (Appendix A and Figure 2), increased pulp ratio was shown to cause significantly higher values of b* and C* in all formulations in comparison to the control. These changes denote increased yellowness and vividness with increased fruit pulp, as shown in Figure 1. In a recent study by Dimitrellou et al. [20], an increase in C* similar to that of the current study was demonstrated in blueberry yogurt formulations.

Overall, there was no significant difference in the values of L*, a*, and h* of all freshly prepared (Appendix A) *Dovyalis caffra* fruit yogurt formulations. As may be further observed in Appendix A, refrigerated storage of 21 days did not have any significant effect on all measured color parameters of *Dovyalis caffra* yogurt formulations. The findings of Dimitrellou et al. [20] in stored grape juice yogurt formulations affirm the results of the present study.

### 3.3. Organoleptic Properties and Their Evolution During Storage of Dovyalis caffra Fruit Yogurt

The organoleptic parameters of freshly prepared *Dovyalis caffra* fruit yogurt formulations are displayed in Figure 3 and Appendix A. As can be clearly observed in Figure 3, the various formulations had an average color score of approximately 7, with no significant difference between individual scores. This implies a moderate liking of color in all formulations and confirms the findings of Matter et al. [11], who reported average color scores of 6.7 and 7.5 in cactus pear and papaya yogurt formulations, respectively. As further shown in Appendix A, storage did not have a significant effect on the color of all formulations. This trend was also reported by Waweru et al. [13] in their study on analogous fruit yogurt formulations constituting papaya pulp and cow milk.

According to Figure 3, a systematic increase of 5% in the pulp ratio of freshly prepared fruit yogurt formulations resulted in significantly lower scores in the taste perception of consecutive products. This could be attributed to increased sourness caused by the high acidity of *Dovyalis caffra* fruit pulp [23]. In general, storage of the various *Dovyalis caffra* fruit yogurt formulations resulted in slightly decreased scores in the taste attribute, although not significantly (Appendix A). This could be attributed to the small increase in acidity during storage, which directly translates to increased sourness. The findings correlate well with those reported for sweet orange marmalade yogurt formulations [26].

From Figure 3, it is clear that an increased pulp ratio leads to significantly lower textural scores of freshly prepared *Dovyalis caffra* fruit yogurt formulations. The general variation of textural scores seen in this study is largely similar to that reported by Matter et al. [11] in their study on papaya and cactus pear yogurt formulations.

With regard to overall acceptability, the control was found to be the most preferred of all yogurt formulations (Figure 3). Formulations D-5 and D-10, however, registered scores above 7, which may be considered fairly acceptable and not significantly different from the control D-0. At 15% pulp inclusion, for D-15, there was, however, a significant reduction in the overall acceptability throughout the storage period (Appendix A). These results are consistent with both the findings of Waweru et al. [13] and Al-Bedrani et al. [26]

### 3.4. Effect of Storage on the Microbial Properties of Dovyalis caffra Fruit Yogurt

Appendix A shows the microbial enumeration of *Dovyalis caffra* fruit yogurt formulations. As evidenced in the table, there was no detection of coliforms, yeast, and molds in all yogurt formulations throughout the storage period. Matter et al. [11] similarly reported zero counts in yeast, molds, and coliforms during a 10-day storage period of papaya and cactus pear yogurt formulations. Contrary to the findings of the current study, however, yeast and molds were reported in strawberry fruit yogurts during the first week of storage and coliforms during the second week [8]. In a general perspective, the non-detection of yeast and molds in *Dovyalis caffra* fruit yogurt is a good indication that pasteurization of the product was effective, while the non-detection of coliforms is a good indication of adherence to hygienic protocols during processing and handling of the product. A previous study by Waweru et al. [13] affirms this concept by first optimizing pasteurization conditions for papaya fruit pulp (based on the elimination of yeast and molds) prior to making fruit yogurt formulations.

## 4. Conclusions and Recommendations

This study investigated the effect of incorporating up to 15% (*w*/*w*) *Dovyalis caffra* fruit pulp into cow milk yogurt. Compared to the control, formulations with added pulp recorded enhanced contents of ascorbic acid, total polyphenols, fiber, total titratable acidity (TTA), and yellowness, which is in line with increasing fruit pulp. This, however, corresponded to significantly lower contents in fat, protein, and viscosity. Pulp incorporation had no significant effect on moisture, ash, and water-holding capacity (WHC) of the formulations. Storage for up to 21 days indicated a significant increase in TTA and a corresponding decrease in pH but no significant change in WHC and viscosity of formulations, indicating a good physical shelf-life of the yogurts. There was no detection of coliforms, yeasts, and molds in all samples throughout the storage period, indicating good microbial shelf-life of the food products.

The organoleptic acceptability of the product was good, up to 10% pulp addition, despite indications that the measured increase in TTA and decreased pH negatively affected taste perception. Increased pulp content, while significantly increasing the yellowness and chroma of the fruit yogurts, did not affect the color scores from the sensory panel, while a decrease in measured viscosity due to pulp addition resulted in poorer texture acceptability of the yogurts. Despite these shortcomings, the overall acceptability of the fruit yogurts remained high (scores of approximately 7.5 out of 9), with up to 10% pulp addition.

Such yogurt formulations also meet the specified requirements in the East African standards for yogurt specification [27] in terms of physicochemical and microbiological quality. These yogurt formulations also pose as potential sources of phytochemicals considering their contents of vitamin C and total polyphenols in comparison to other dairy products.

Despite the demonstrated potential of *Dovyalis caffra* fruit in yogurt formulation, there is still a need for further considerations. For instance, studies on possible ways of regulating the fruit’s high acidity should be conducted in order to facilitate the incorporation of the fruit components into yogurts in higher proportions. Further diversified research on new food product development from *Dovyalis caffra* fruits is also highly encouraged.

## Figures and Tables

**Figure 1 foods-13-04102-f001:**
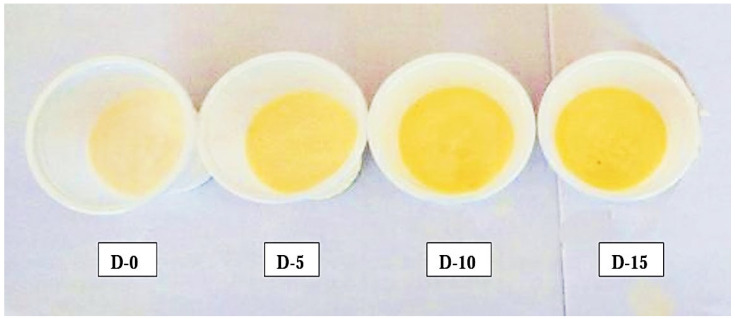
*Dovyalis caffra* fruit yogurt formulations: [D-0] = 0% pulp (plain yogurt); [D-5] = 5% pulp; [D-10] = 10% pulp; and [D-15] = 15% pulp.

**Figure 2 foods-13-04102-f002:**
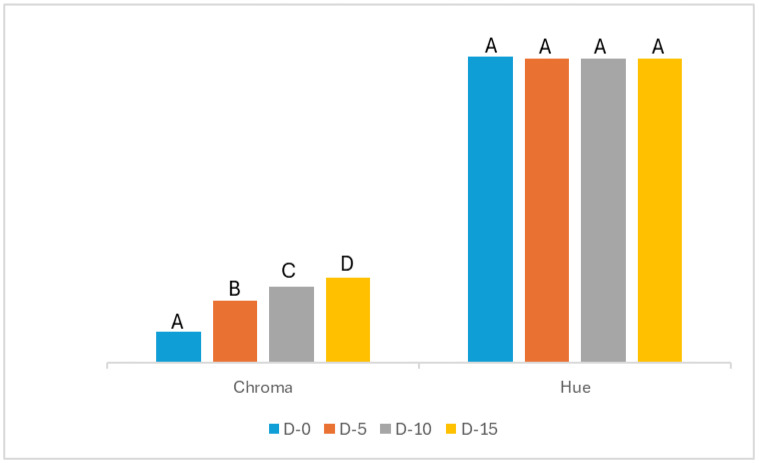
Color parameters, chroma, and hue of yogurt formulations on day 1: [D-0] = 0% pulp (plain yogurt); [D-5] = 5% pulp; [D-10] = 10% pulp; and [D-15] = 15% pulp. Means in columns within a specific parameter with the same upper case superscript are not significantly different (*p* > 0.05).

**Figure 3 foods-13-04102-f003:**
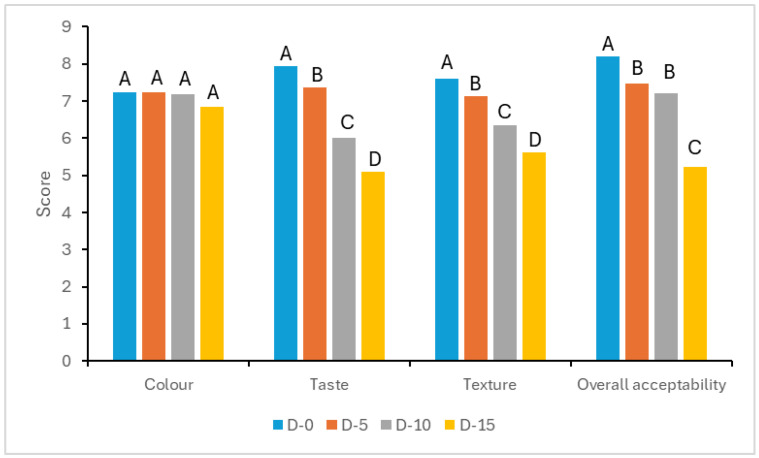
Organoleptic properties of freshly prepared *Dovyalis caffra* fruit yogurt formulations: [D-0] = 0% fruit pulp (Plain milk yogurt); [D-5] = 5% fruit pulp; [D-10] = 10% fruit pulp; and [D-15] = 15% fruit pulp. Means in columns within a specific parameter with the same upper case superscript are not significantly different (*p* > 0.05).

**Table 1 foods-13-04102-t001:** Basic chemical constituents of fresh *Dovyalis caffra* fruit yogurt (1-day-old).

Product	Parameter							
	Moisture (g/100 g)	Protein (g/100 g)	Fat (g/100 g)	Ash (g/100 g)	Fiber (g/100 g)	CHO (g/100 g)	Ascorbic Acid (mg/100 g)	Total Polyphenols (mg GAE/100 g)
D-0	85.09 ± 0.02 ^a^	4.35 ± 0.06 ^c^	2.30 ± 0.02 ^b^	0.53 ± 0.02 ^a^	ND	7.79 ± 0.01 ^a^	0.02 ± 0.01 ^a^	1.84 ± 0.12 ^a^
D-5	85.21 ± 0.22 ^a^	4.12 ± 0.01 ^bc^	2.21 ± 0.01 ^ab^	0.50 ± 0.02 ^a^	0.02 ± 0.00 ^a^	8.11 ± 0.01 ^b^	5.38 ± 0.08 ^b^	6.67 ± 0.08 ^b^
D-10	85.10 ± 0.02 ^a^	3.94 ± 0.04 ^ab^	2.15 ± 0.08 ^ab^	0.48 ± 0.01 ^a^	0.04 ± 0.01 ^ab^	8.41 ± 0.01 ^c^	10.75 ± 1.01 ^c^	13.37 ± 0.25 ^c^
D-15	85.17 ± 0.02 ^a^	3.81 ± 0.13 ^a^	2.02 ± 0.05 ^a^	0.47 ± 0.01 ^a^	0.07 ± 0.02 ^b^	8.63 ± 0.01 ^d^	14.49 ± 0.68 ^d^	17.02 ± 0.15 ^d^

Data presented as means ± standard deviation, Means in columns with the same superscript are not significantly different (*p* > 0.05), CHO = Carbohydrate, ND = Not Detected.

**Table 2 foods-13-04102-t002:** Effect of storage on the physicochemical parameters of *Dovyalis caffra* fruit yogurt.

Parameter	Product	Storage Period			
		Day 1	Day 7	Day 14	Day 21
pH	D-0	4.21 ± 0.02 ^D,c^	4.20 ± 0.02 ^c^	3.86 ± 0.01 ^b^	3.64 ± 0.02 ^a^
	D-5	3.84 ± 0.04 ^C,c^	3.84 ± 0.02 ^c^	3.51 ± 0.01 ^b^	3.28 ± 0.02 ^a^
	D-10	3.52 ± 0.03 ^B,c^	3.51 ± 0.04 ^c^	3.26 ± 0.02 ^b^	3.06 ± 0.01 ^a^
	D-15	2.97 ± 0.01 ^A,c^	2.93 ± 0.03 ^bc^	2.90 ± 0.01 ^ab^	2.85 ± 0.02 ^a^
TTA (%)	D-0	1.32 ± 0.04 ^A,a^	1.32 ± 0.01 ^a^	1.42 ± 0.02 ^b^	1.51 ± 0.01 ^b^
	D-5	1.68 ± 0.03 ^B,a^	1.67 ± 0.01 ^a^	1.81 ± 0.02 ^b^	1.93 ± 0.02 ^c^
	D-10	1.81 ± 0.01 ^C,a^	1.82 ± 0.01 ^a^	1.97 ± 0.02 ^b^	2.13 ± 0.04 ^c^
	D-15	2.02 ± 0.04 ^D,a^	2.08 ± 0.00 ^a^	2.23 ± 0.02 ^b^	2.36 ± 0.01 ^c^
WHC (%)	D-0	73.30 ± 0.75 ^A,a^	73.13 ± 0.52 ^a^	72.46 ± 0.99 ^a^	72.30 ± 0.75 ^a^
	D-5	72.85 ± 0.83 ^A,a^	72.68 ± 0.60 ^a^	72.02 ± 1.07 ^a^	71.85 ± 0.83 ^a^
	D-10	72.33 ± 0.51 ^A,a^	71.96 ± 0.55 ^a^	71.30 ± 1.03 ^a^	71.13 ± 0.79 ^a^
	D-15	74.21 ± 1.12 ^A,a^	74.54 ± 0.17 ^a^	73.88 ± 0.64 ^a^	73.21 ± 1.12 ^a^
Viscosity (Pa·s)	D-0	0.40 ± 0.04 ^A,a^	0.41 ± 0.05 ^a^	0.40 ± 0.02 ^a^	0.40 ± 0.04 ^a^
	D-5	0.33 ± 0.05 ^A,a^	0.32 ± 0.03 ^a^	0.32 ± 0.04 ^a^	0.32 ± 0.04 ^a^
	D-10	0.32 ± 0.04 ^A,a^	0.31 ± 0.03 ^a^	0.32 ± 0.04 ^a^	0.31 ± 0.03 ^a^
	D-15	0.31 ± 0.03 ^A,a^	0.30 ± 0.02 ^a^	0.30 ± 0.02 ^a^	0.30 ± 0.02 ^a^

Data presented as means ± standard deviation. Means in columns within a specific parameter with the same upper case superscript are not significantly different (*p* > 0.05). Means in rows with the same lowercase superscript are not significantly different (*p* > 0.05).

## Data Availability

The raw data supporting the conclusions of this article will be made available by the authors on request.

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
