# Peer review of "Effect of Different Formulations and Storage on the Physicochemical, Microbiological, and Organoleptic Characteristics of *Dovyalis caffra* Fruit Yogurt"

_foods, 2024, doi:10.3390/foods13244102_

Round 1

Reviewer 1 Report

Comments and Suggestions for Authors

This study examined the effect of adding 5%, 10% and 15% of Dovyalis caffra fruit pulp to cow milk yoghurt. Its physicochemical, microbiological, and sensory properties were analysed over 21 days of refrigerated storage. The manuscript is well-written, but it lacks the scientific depth. The results are predictable, and the relationship between various parameters was not discussed. 

Introduction:

The introduction section is too long, with individual papers discussed in each paragraph. The Authors should have combined similar results obtained by different authors in one sentence with multiple references.

Material and Methods:

The experimental design is very basic, with no consideration of the potential relationship between determined parameters.

The procedure for mixing Dovyalis caffra fruit pulp with yoghurt was not described, and it can have a significant influence on yoghurt viscosity.

The proximate analysis section is written with too many details. The Authors used standard and well-known methods for the determination of moisture, protein, fat, ash and crude fibre content, and therefore they should be stated only by the AOAC method number. The methods for ascorbic acid and total phenolic content should be described in individual sections, separately from proximate analysis.

Sensory analysis was not described properly. Namely, it is not clear whether trained or untrained panellists evaluated yoghurt samples.

Results and Discussion:

There is no in-depth discussion on the relationship between nutritional composition and physicochemical properties of yoghurts, as well as on their influence on sensory acceptance.

The cause of lower viscosity observed in yoghurts with the addition of Dovyalis caffra fruit pulp was not discussed properly – could it be due to the vigorous mixing?

The performed sensory analysis is too simple to determine the origin of the low acceptability of yoghurts with higher levels of Dovyalis caffra fruit pulp.

The antioxidant properties of yoghurts were not analysed, despite the claim that these products have functional properties.

Author Response

Thank you very much for taking the time to review this manuscript. Please find the detailed responses below and the corresponding revisions/corrections highlighted in blue in the re-submitted files.

Reviewer 2 Report

Comments and Suggestions for Authors

Review on manuscript: foods-3307437

Effect of different formulations and storage on the physico-chemical, microbiological and organoleptic characteristics of Dovyalis caffra fruit yoghurt

by Daniel Mwangi Waweru, Joshua Mbaabu Arimi, Eunice Marete, Niamh Harbourne and Jean-Christophe Jacquier

submitted to Foods

In the manuscript submitted for review, the authors studied the effect of Dovyalis caffra fruit on the properties of yoghurt.

The topic taken up by the authors is not new, and the only novelty is the use of specific fruits. The methodology is incorrectly presented.

Detailed recommendation:

line 19 – why lower score but not lower fat content?

line 19 – should be: apparent viscosity,

line 34 – fruits are not a good source of proteins, i.e. amino acids,

line 104 – why foodstuffs and not just yogurt?

line 122 – what exactly does room temperature mean?

Methods – the description of methods, especially those included in standards, should not be a laboratory procedure, therefore the entire methodology section should be significantly shortened and the most important necessary information should be left,

line 184 – why crude fiber and not dietary fiber?

line 245 – concentration should be expressed as molarity,

line 251 – instead of rpm, the centrifugal force should be specified,

line 264 – type of illuminant and measurement geometry should be provided,

lines 273-274 – equations are not needed,

line 289 – were the panelists selected and trained?

line 306 – what does lower score mean here?

line 340 – should be: apparent viscosity,

Figure 2 – no Y axis description,

lines 372-373 – the authors should decide whether they performed an organoleptic or sensory evaluation, as these are not the same concepts,

Figure 3 – incorrect Y axis description,

References – should be checked and corrected, in many places pages or article numbers are missing.

Comments on the Quality of English Language

see report 

Author Response

(The authors gave the same response as above.)

Reviewer 3 Report

Comments and Suggestions for Authors

The manuscript presents an interesting study; however, several revisions are recommended to enhance clarity, scientific rigor, and overall quality. Below are specific suggestions organized by section. While the research topic and methodology are promising, certain aspects require more comprehensive details. Therefore, I recommend minor revisions.

(a)    Consider including a more detailed discussion of relevant previous studies to better position this work within the existing body of research. Emphasizing how this study addresses specific gaps in the literature would significantly strengthen the manuscript’s impact.

(b)     Presenting the experimental design in a table, rather than as text, would improve clarity and help readers navigate the setup more easily.

(c)    Please specify exact values or ranges for key parameters such as temperature and pH at each stage of preparation, processing, and storage to ensure that the methodology is fully transparent. While the addition of pulp is noted, clarify the specific ratios of all ingredients such as milk, fruit pulp, and additives. For instance, provide the initial fat, protein, and sugar content of the milk used and explain the choice of the 5%, 10%, and 15% concentrations of Dovyalis caffra pulp.

(d)    Specify the quantity, concentration, and strains of Lactobacillus bulgaricus and Streptococcus thermophilus added, as these can affect the texture, flavor, and fermentation profile of the yoghurt. Add a clear sequence of steps in yoghurt preparation, including exact times and temperatures for pasteurization, inoculation, and incubation, in the supplementary material to ensure replicability.

(e)    Specify the brand, model, and settings for key equipment such as the pasteurizer, mixer, and rheometer; for example, when discussing viscosity testing with a rheometer, mention the model, shear rate, temperature, and duration of measurement to allow for accurate replication.

(a)    Explicitly state the sample sizes and the number of replicates for each measurement, which is important for assessing the robustness and statistical power of the analysis. Also specify the software used for statistical analysis.

(b)    Did you use control samples? If control samples were used, please describe their composition in detail.

Author Response

(The authors gave the same response as above.)

Round 2

Reviewer 1 Report

Comments and Suggestions for Authors

The authors have significantly improved the manuscript. However, there are still minor issues that should be corrected before the paper is published. Regarding the quality of the English language, Total Phenolic Content. Gallic acid, Chroma and Hue should be written in lowercase throughout the manuscript.

Introduction:

The Authors did not make any changes to the Introduction section, which is still too long. Taking into account the authors' explanation for this, I suggest deleting the sentences about adding marmalades (Lines 56-60) and juices (Lines 67-72 and Lines 83-86) to yoghurt since this paper is about adding fruit pulp. Namely, marmalade and juices have a consistency that differs greatly from fruit pulp; therefore, these results cannot be compared.

Material and Methods:

Since untrained panellists evaluated yoghurt samples using a nine-point hedonic scale, the performed analysis should be called “Consumer preference testing”, according to ISO 11136:2014 (Sensory analysis – Methodology – General guidance for conducting hedonic tests with consumers in a controlled area). This change should be made throughout the manuscript, including headings “2.8. Sensory analysis” and “3.3. Sensory properties and their evolution during storage of Dovyalis caffra fruit yoghurt”.

Results and Discussion:

The authors said that it can be hypothesized that the fruit pulp addition weakens the milk protein gel structure. Since the addition of Dovyalis caffra fruit pulp increased the dietary fibre content of yoghurts, it can be supposed that some interaction between dietary fibre and milk protein gel structure occurs. This should be discussed taking into account previous studies on this type of interaction in fermented dairy products.

Author Response

Thank you very much for taking the time to review this revised manuscript. Please find the detailed responses below and the corresponding revisions/corrections highlighted in red in the re-submitted files
